# Isolation and Identification of Chromium Reducing *Bacillus Cereus* Species from Chromium-Contaminated Soil for the Biological Detoxification of Chromium

**DOI:** 10.3390/ijerph17062118

**Published:** 2020-03-23

**Authors:** Ming-hao Li, Xue-yan Gao, Can Li, Chun-long Yang, Chang-ai Fu, Jie Liu, Rui Wang, Lin-xu Chen, Jian-qiang Lin, Xiang-mei Liu, Jian-qun Lin, Xin Pang

**Affiliations:** State Key Laboratory of Microbial Technology, Shandong University, No. 72 Binhai Road, Qingdao 266237, China; liminghao.777@163.com (M.-h.L.); 201411725@mail.sdu.edu.cn (X.-y.G.); lican0826@163.com (C.L.); 201311646@mail.sdu.edu.cn (C.-l.Y.); 201812484@mail.sdu.edu.cn (C.-a.F.); 201732224@mail.sdu.edu.cn (J.L.); rwang@sdu.edu.cn (R.W.); linxuchen@sdu.edu.cn (L.-x.C.); liuxiangmei@sdu.edu.cn (X.-m.L.)

**Keywords:** heavy metal contamination, chromium (VI), bioreduction, bioremediation, *Bacillus cereus*, immobilized cells

## Abstract

Chromium contamination has been an increasing threat to the environment and to human health. Cr(VI) and Cr(III) are the most common states of chromium. However, compared with Cr(III), Cr(VI) is more toxic and more easily absorbed, therefore, it is more harmful to human beings. Thus, the conversion of toxic Cr(VI) into Cr(III) is an accepted strategy for chromium detoxification. Here, we isolated two *Bacillus cereus* strains with a high chromium tolerance and reduction ability, named *B. cereus* D and 332, respectively. Both strains demonstrated a strong pH and temperature adaptability and survival under 8 mM Cr(VI). *B. cereus* D achieved 87.8% Cr(VI) removal in 24 h with an initial 2 mM Cr(VI). Cu(II) was found to increase the removal rate of Cr(VI) significantly. With the addition of 0.4 mM Cu(II), 99.9% of Cr(VI) in the culture was removed by *B. cereus* 332 in 24 h. This is the highest removal efficiency in the literature that we have seen to date. The immobilization experiments found that sodium alginate with diatomite was the better method for immobilization and *B. cereus 332* was more efficient in immobilized cells. Our research provided valuable information and new, highly effective strains for the bioremediation of chromium pollution.

## 1. Introduction

Chromium is a toxic heavy metal contaminant [1,2,3,4], which is heavily produced in waste from industries, including tanneries, electroplating, metal smelting, acid mine drainage, and other industries [5,6,7]. These untreated chromium-containing pollutants are mostly released into the environment, posing threats to ecosystems [2,5]. There are two common and stable oxidation states of chromium in the environment, namely Cr(VI) and Cr(III) [8,9,10]. Cr(III) is involved in sugar and lipid metabolism and is less toxic to humans [8,11,12], while Cr(VI) is carcinogenic, teratogenic, and mutagenic and is easily absorbed by humans through biomagnification in the food chain [13,14]. The permissible limit concentration for Cr(VI) in drinking water is less than 50 μg/L, as recommended by the World Health Organization (WHO) and United States Environmental Protection Agency (USEPA) [15,16,17]. In contrast, the first level standard of chromium content in a soil environment is less than 90 mg/kg. However, the chromium content of industrial wastes far exceeds these safety standards, and could be as high as 717.61 mg/kg in tannery sludge [5].

Due to the toxicity of this metal, chromium-containing contaminants must be detoxified before being released into the environment [15,18]. Conventional strategies used for chromium removal include chemical reduction methods, precipitation, ion exchange, and electro-dialysis [16,17,19,20]. However, these traditional methods are costly and difficult to apply in treating contaminated soil [16,19,21]. Chemical reduction and precipitation methods, on the other hand, usually require the addition of other chemicals, thus resulting in potential secondary pollution [20,22,23]. Therefore, it is important to develop low-cost and ecofriendly in situ chromium removal methods [23,24]. 

Since the discovery of chromium-reducing bacteria, the use of microorganisms to reduce Cr(III) to the less toxic Cr(VI) has gained more attention [10,20]. Many species have been described to have the ability to reduce hexavalent chromium [20,21,25], such as *Acidithiobacillus* [26], *Acinetobacter* [27], *Arthrobacter* [28], *Achromobacter*, *Bacillus* [15], *Brucella* [20], *Desulfovibrio* [29], *Enterobacter* [30], *Leucobacter* [31], *Mirococcus* [32], *Ochrobactrum* [32], *Pseudomonas* [32] and *Thermus* [33]. However, chromium-contaminated soils and waters are usually not conducive to the growth and reproduction of these microorganisms. Therefore, the in situ reduction in Cr(VI) requires a species with strong growth and high chromium reducing activity [22,23].

Here, we isolated and identified two chromium tolerance and reduction efficiency *Bacillus cereus* strains, named *B. cereus* D and 332, respectively. The two strains can both survive under 8 mM Cr(VI), making them advantageous in treating chromium-contaminated soil and water. The factors affecting the growth and chromium reduction, including temperature and pH, were determined. Cu(II) was found to increase the removal rate of Cr(VI). Meanwhile, a more effective immobilization method, using sodium alginate and diatomite, was determined in this study. Our results demonstrated that *B. cereus* D had a higher removal rate of Cr(VI) in planktonic cells, while *B. cereus* 332 was more efficient in immobilized cells. Our research provided new, highly effective strains for the biological detoxification of chromium.

## 2. Materials and Methods

### 2.1. Growth Conditions

Bacteria were grown in Luria-Bertani (LB) medium, consisting of 10 g/L peptone, 5 g/L yeast extracts, and 10 g/L NaCl. The solid LB medium was added with 1.5% (*w*/*v*) agar [34]. All the media were autoclaved at 121 °C for 20 min.

### 2.2. Isolation of the Strains

The strains were isolated from the soil collected near electroplating plants from Shandong, Jiangxi, Jilin, Liaoning, and Heilongjiang provinces. Briefly, 1 g of soil of each sample was dissolved in 10 mL sterile water, and then 100 μL of this soil suspension was added to LB medium with 2 mM Cr(VI) [15,35]. After incubating for 72 h at 37 °C, with agitation at 180 rpm, the enriched culture solution was then spread-plated on LB-solid medium with 2 mM Cr(VI) [35]. The selected clones were further purified using the streak plating method [36]. The chromium reduction ability of the initially screened strains was tested (see Section 2.5) and the two strains with the highest activity of chromium reduction were selected for further experiments.

### 2.3. Morphological Observation of Strains

Bacterial morphology was characterized by scanning electron microscopy [37,38,39]. Cells were prepared from overnight stationary cultures, and the samples were fixed with 2.5% glutaraldehyde for 4 h at 4 °C, and centrifuged at 5000× *g* for 1 min. The pellets were sequentially dehydrated in a gradient of 30%, 50%, 70%, 80%, 90%, 95%, and 100% ethanol for 20 min each and then dried at the critical point with CO_2_ [38,39]. After sputtering with gold, the morphology and the size of bacteria were then obtained using scanning electron microscopy (Quanta 250 FEG, FEI, Hillsboro, Oregon, USA) [37,38].

Gram staining was performed as described in the Laboratory Manual and Workbook in Microbiology [40]. First, the cells were fixed on glass slides, then stained with ammonium oxalate crystal violet for 1 min, followed by mordant dyeing with iodine for 1 min. The cells were then decolorized with 95% ethanol for 20 s and counterstained with safranine for 2 min. The stained cells were then visualized under an upright microscope (Olympus BX51T, Kyoto, Japan).

Spore staining was conducted to determine whether the isolated bacteria were able to produce spores. The cells were dyed with 5% malachite green for 15 min at room temperature and counterstained with safranine for 2 min [41] before being observed under the microscope.

Physiological and biochemical characterization, including the MR-VP test, motive test, lysozyme test, nitrate test, lecithin test, acid production, hemolysis test, protein crystal test, root shape growth, and casein decomposition, were carried out using physiological and biochemical test kits from Hopebio (Qingdao, China) following the manufacturer’s instructions.

### 2.4. Phylogenetic Analysis Using 16S rDNA

Genomic DNA was extracted using the TIANamp Bacteria DNA Kit from Tiangen (Beijing, China), which was then used to amplify the genes of 16S rDNA with PCR, using the primer pairs 16S-F/16S-R (16S-F: AGAGTTTGATCCTGGCTCA; 16S-R: GGTTACCTTGTTACGACTT) [42]. The amplicons were purified after separating by agarose gel using a Gel Extraction Kit (Omega, Shanghai, China) and were sent to GENEWIZ (Jiangsu, China) for sequencing. The 16S rDNA sequences of the strains isolated were deposited in NCBI (https://www.ncbi.nlm.nih.gov/) under the GenBank Accession Numbers MK071611 (*B. cereus* D) and MK071612 (*B. cereus* 332). The generated sequences were searched for most related sequences in NCBI (https://www.ncbi.nlm.nih.gov/) using the BLASTn function, and these related sequences were eventually used to generate phylogenetic trees in MEGA 7.0.

### 2.5. Determining Cr(Ⅵ) in the Culture Medium

The concentration of the Cr(VI) in the culture media was measured following the diphenyl carbazide (DPC) method [43,44,45,46]. Briefly, 200 μL samples were prepared and mixed with a reaction solution containing 300 μL 0.6% diphenyl carbazide and 1000 μL acid buffer (25% sulfuric acid and 25% phosphate) [43,45], then ddH_2_O was added up to 25 mL. The reaction was allowed to stand at room temperature for 2 min, then, the absorbance of the reaction solution was measured at 540 nm using UV-Vis 1800 (Shimadiu, Kyoto, Japan) [15].

### 2.6. Conditions for Bacterial Growth

The cultures were initially incubated overnight and collected by centrifugation at 10,000× *g* for 5 min at 4 °C. The bacterial concentrations were then adjusted to OD_600 nm_ = 1 before inoculation [47], and only 0.1% of the adjusted bacterial suspension was inoculated to a fresh medium.

The effect of potassium dichromate on the growth of the isolated strains was also tested in a 100-well bacterial culture plate. Three replicates for each potassium dichromate concentration (1, 2, 4, 6, and 8 mM) were used with the pH of the media maintained at 7.0. After incubation at 37 °C for 24 h at 180 rpm, the biomass of the bacteria was measured at OD_600 nm_ using automatic growth curve analyzer (BioscreenC, Finland) [43,44].

The effect of pH (3, 4, 5, 6, 7, 8, 9, and 10) on the growth of the isolated strains was measured in a 100-well bacterial culture plate. The pH was adjusted by adding either 0.1 M HCl or 0.1 M NaOH as needed, which were tested in triplicate. After inoculation, cultures were incubated at 37 °C for 24 h at 180 rpm with the biomass measured at OD_600 nm_ using automatic growth curve analyzer (BioscreenC, Finland).

The effects of temperature (25, 30, 35, 37, and 40 °C) on the growth and chromium reduction in the strains were measured and incubated a 48-well bacterial culture plate at 180 rpm, pH = 7.0 for 24 h. The biomass of the bacteria was measured at OD_600 nm,_ and the reductions in the chromium were determined using the diphenyl carbazide (DPC) method, as described.

The shake flask experiments including the growth curve and chromium reduction in the strains were measured and incubated in 100 mL flasks at 2 mM Cr(VI), 180 rpm, pH = 7.0 for 24 h, with sampling every three hours for cell growth and chromium reduction determination. The biomass of the bacteria was measured at OD_600 nm_ and the reductions of the chromium were determined using the diphenyl carbazide (DPC) method as formerly described.

The effect of the chemical factors on the growth of the isolated strains was investigated in a 48-well bacterial culture plate. A total of 0.4 mM of Cu(II), Ni(II), Zn(II), Mn(II), and Ca(II) and 1 mM glucose (Glc), fructose (Fru), glycine (Gly), and trisodium citrate dihydrate (Tcd) were chosen as the test chemicals and control experiments were set without these chemicals added. After incubation at 37 °C for 24 h at 180 rpm, the reduction in the chromium was determined using the diphenyl carbazide (DPC) method as formerly described.

### 2.7. Immobilization of the Bacteria

The cells were immobilized as described previously [48,49,50,51], where mid-log phase cells were collected by centrifugation at 10,000× *g* for 5 min at 4 °C and adjusted to OD_600 nm_ = 1 with sterile water [47]. We then collected 10 mL of the adjusted bacterial suspension and resuspended the collected suspension in 1 mL saline solution. After this, 10 mL of the embedding medium was added to the samples. The immobilized beads were prepared by embedding the medium with bacteria, which was slowly added to the crosslinking agent (2% CaCl_2_ and 4% H_3_BO_3_) with a syringe [50,51]. Three working solutions of the embedding media were used in this study. The first included a mixture of 2% sodium alginate (SA) and 4% polyvinyl alcohol (PVA) [48], the second included only 2% sodium alginate, and the third included a mix of 2% sodium alginate and 1% diatomite (DIA) [49].

### 2.8. Statistical Analysis

All experiments were performed three times and three biological replicates were set for all experimental samples. Student’s *t*-test, using a GraphPad Prism (version 7.0), was used to conduct a statistical analysis. Additionally, **** indicates *p* < 0.0001, *** indicates *p* < 0.001, ** indicates *p* < 0.01, and * indicates *p* < 0.05.

## 3. Results and Discussion

### 3.1. Isolation and Identification of Strains

In this study, two bacterial strains with high chromium tolerance and reduction efficiency were isolated from chromium-contaminated soil, named strains D and 322. Both strains were rod-shaped, Gram-positive, with spores, and the sizes of the D and 332 strains were, respectively, 2.2–3.4 × 0.6–0.8 μm and 4.3–8.4 × 0.8–1.0 μm (Figure 1). The complete details of the biochemical and physiological characteristics of the two strains are summarized in Table 1. The physiological and biochemical characteristics showed that the two isolates demonstrated characteristics of *Bacillus*, such as positive MR-VP, lysozyme, and nitrate tests. These results indicated that the two isolated strains were putatively identified as *Bacillus* based on Bergey’s Manual of Systematic Bacteriology [52]. Phylogenetic analysis of the 16S rDNA gene further revealed that the two isolated strains were *Bacillus cereus* (Figure 2).

### 3.2. Chromium Tolerance of the Isolated Strains

To explore the chromium tolerance of the two isolated *B. cereus* strains, potassium dichromate was added to the culture to simulate the chromium-contaminated environment. From Figure 3, we can see that the two strains showed amazing chromium tolerance and could survive under 8 mM Cr(VI). The growth of the two strains was similar without the addition of Cr(VI). However, the growth of *B. cereus* D was better than that of *B. cereus* 332 under 1, 2, 4, and 6 mM Cr(VI) stress, reflecting the stronger chromium tolerance of *B. cereus* 332. Considering the biomass and the chromium toxic affect, the initial concentration of 2 mM Cr(VI) was used for subsequent research experiments according to Figure 3. Many chromium-resistant strains have been reported previously. For example, a *B. cereus* isolated by Zhao et al. (2012) grows well in 3.8 mM Cr(VI); however, its reducing capacity for Cr(VI) significantly decreased under high concentrations of chromium [25]. The other *B. cereus* isolated by Murugavelh and Mohanty (2013) could also grow in 1.34 mM Cr(VI) and the initial concentration of Cr(VI) was 1.15 mM in other experiments [15]. The *Pseudomonas gessardii* strain LZ-E, isolated from wastewater, could only tolerate 0.7 mM Cr(VI) [23]. Based on these previous studies, the two strains *B. cereus* D and *B. cereus* 332 in this study demonstrated a higher tolerance to Cr(VI), suggesting their stronger potential in reducing chromium. Thus, it is valuable to further study the growth and chromium reduction ability of these two strains.

### 3.3. The Effect of pH on the Growth of the Isolated Strains

pH is an important factor affecting the growth of bacteria. We measured the growth of the bacteria at different pH levels from 3 to 10 (Figure 4). The results demonstrated that pH 7.0 is the most suitable environment for the growth of the *B. cereus D* (Figure 4A). *B. cereus* D tolerated a pH range from 6–10 and an acidic environment (pH < 5) was not conducive for growth (Figure 4A). As for *B. cereus* 332, the highest biomass was obtained at pH 6.0 (Figure 4B). *B. cereus* 332 tolerated a pH range from 5–10. Both *B. cereus* D. and *B. cereus* 332 were able to tolerate changes in the pH of the external environment, indicating their great potential for applications in pollution control. The pH value of chromium-containing wastewater or sludge is usually neutral [5,26]. Therefore, pH 7.0 was selected as the culture condition for subsequent experiments.

### 3.4. The Effect of Temperature on the Growth and Chromium Reduction of the Isolated Strains

Temperature is an essential factor affecting the production cost of industrial applications. It is also a key element affecting the growth of bacteria and the enzyme activity. The biomass and the chromium reducing capacity of the two isolated strains in response to five temperatures (25, 30, 35, 37, and 40 °C) were determined (Figure 5). The two isolated strains demonstrated a high adaptability to different temperatures from 25 to 40 °C. Both *B. cereus* D and *B. cereus* 332 obtained the maximum biomass at 30 °C (Figure 5A). Thus, the optimum temperature for the growth of both *B. cereus* D and *B. cereus* 322 was 30 °C (Figure 5A). 

The reduction in Cr(VI) was determined under different temperature conditions with an initial 2 mM Cr(VI) concentration at pH 7.0 (Figure 5B). The chromium-reducing ability of *B. cereus* D increased with the temperature, reaching as high as a 66.4% reduction in Cr(VI) at 40 °C within 24 h. The removal rate of Cr(VI) in *B. cereus* D was 60.2% at 37 °C within 24 h (Figure 5B). A higher temperature would also mean increased costs in industrial applications. Thus, considering the costs and reduction efficiency, 37 °C was selected as the experimental temperature for chromium reduction for *B. cereus* D. Additionally, the highest Cr(VI) removal rate of *B. cereus* 332 was obtained at 37 °C with a value of 57.4% within 24 h. This was consistent with previous reports showing that the optimum temperature for other *B. cereus* isolates to reduce Cr(VI) was also 37 °C [15,25].

### 3.5. The Reducing Ability of the Isolated Strains

The shake flask experiments are the beginning of the industrial application of the strains. The growth and chromium-reducing capacity of the two isolated strains was monitored with or without 2 mM Cr(VI) in 100 mL shake flasks at 37 °C, pH = 7.0 (Figure 6). The growth of *B. cereus* 332 was slightly higher than that of *B. cereus* D without the addition of Cr(VI) (Figure 6A). The presence of chromium was detrimental to the organisms, causing slower growth and a decreased biomass. This adverse effect was more pronounced for *B. cereus* 332, which eventually caused the growth of *B. cereus* 332 to be significantly lower than that of *B. cereus* D under 2 mM Cr(VI) stress (Figure 6B). For the chromium-reducing ability, we can see from Figure 6C that the Cr(VI) was also gradually reduced with the growth of the bacteria. After 24 h, 87.8% and 61.0% of Cr(VI) was already reduced by *B. cereus* D and *B. cereus* 332, respectively (Figure 6C). *B. cereus* D showed a higher chromium reduction efficiency and ability than *B. cereus* 332 (Figure 6C).

*B. cereus* isolated by Murugavelh et al. removed 96.8% of Cr(VI) in 48 h with a 1.15 mM initial Cr(VI) concentration [15]. Another article reported that the removal rate of Cr(VI) was 82% after 72 h with a 2 mM initial Cr(VI) concentration in mixed anaerobic culture [22]. By comparing the experimental data, the bacteria isolated in this study demonstrated a higher chromium reduction efficiency, not only in the initial chromium reduction concentration but also in the chromium reduction time. Furthermore, the leaching method has been mostly used as a strategy for the in situ treatment of polluted soils [53,54,55,56], where the chromate in the soil was continuously eluted by the eluent, typically using the bacteria as the eluent. For example, previous reports demonstrated the use of the bacteria *Acidithiobacillus ferrooxidans* as the eluent to elute chromium from soil; however, this took as long as 5 weeks [57]. Our results demonstrated that the *B. cereus* isolated in this study had a stronger chromium reduction ability in the planktonic cell and had more potential for application in the treatment of chromium-contaminated soil through bacterial suspension spraying.

### 3.6. The Effect of Chemical Factors on the Ability of B. Cereus D and 332 in Reducing Chromium

In order to further improve the chromium reduction efficiency of the strains, some chemical factors were used in our study. Five metal ions, including Cu(II), Zn(II), Ni(II), Ca(II), and Mg(II), and four organic molecules, including glucose (Glc), fructose (Fru), glycine (Gly), and trisodium citrate dihydrate (Tcd), were added to detect the effect of these chemical factors on the chromium reducing ability of the isolated strains (Figure 7). The efficiency of the two isolated strains in reducing chromium significantly increased in the presence of 0.4 mM copper ions (Figure 7). The removal rate of Cr(VI) was 93.1% within 24 h for *B. cereus* D and as high as 99.9% for *B. cereus* 332 within 24 h under the initial 2 mM Cr(VI) with the addition of Cu(II). The data demonstrated that the addition of a small amount of Cu(II) resulted in a huge chromium reduction in efficiency. It was first reported in 2003 that Cu(II) could actually stimulate the reduction in Cr(VI) using the cell-free extracts of *Bacillus sp*. ES 29 [58]. In 2006, Elangovan et al. also found that the chromate reductase was stimulated by the presence of Cu(II) [59]. Cu(II) was a prosthetic group of many reductases, involved in electron transport [58]. However, the underlying mechanisms involved in the increased ability for chromium reduction under Cu(II) stimulation remain unclear.

### 3.7. Chromium Reduction of the Immobilized Strains

Immobilized cell technologies provide a clear advantage in industrial applications [48,51]. This technology increased the resistance of the cell to the adverse environment, where the immobilized cell can also be repeatedly used [48,51]. The isolated strains *B. cereus* D and 332 were immobilized by traditional methods, specifically in beads (Figure 8A). Different formulations were applied in the immobilization, which affected the level of chromium reduction in the isolated strains. As shown in Figure 8B, the strongest reduction was achieved when sodium alginate was used to immobilize the *B. cereus* D. Using sodium alginate and sodium alginate with diatomite, respectively, 66.9% and 61.3% of Cr(VI) were reduced by the immobilized beads from *B. cereus* D in 120 h. However, immobilized beads using sodium alginate are not hard and easily broken. The immobilized beads from *B. cereus* 332 reduced the chromium in a shorter time. The immobilized beads of *B. cereus* 332 using sodium alginate with diatomite achieved a chromium removal rate of 88.9% in 72 hours. Diatomite was reported to increase the hardness of the immobilized beads [49]. These results indicated that sodium alginate with diatomite was the better method for immobilization. The experimental data demonstrated that *B. cereus* 332 had more potential for immobilization applications due to its higher chromium reducing capability.

## 4. Conclusions

In this study, we isolated two strains from the chromium-contaminated soil and identified them as *B. cereus* through physiological and biochemical experiments and 16S rDNA sequencing. Both strains demonstrated resilient pH and temperature adaptability and survival at a high chromium concentration of 8 mM, showing their high chromium tolerance and application potential. In the planktonic cell state, the *B. cereus* D strain achieved 87.8% Cr(VI) removal in 24 h with an initial 2 mM Cr(VI) concentration at 37 °C at pH 7.0, and the growth and chromium removal ability were higher than those of *B. cereus* 332. The Cu(II) significantly improved the chromium reduction ability of the strains. Almost all the Cr(VI) in the culture environment were removed by *B. cereus* 332 in 24 h with the addition of 0.4 mM Cu(II), and the removal efficiency of chromium was as high as 99.9%. The immobilization experiments found that sodium alginate with diatomite was the better method for immobilization, and *B. cereus* 332 was more efficient in immobilized cells. Our study isolated two *B. cereus* strains with a high application value for the bioremediation of chromium pollution and provided efficient strains for the biological detoxification of chromium.

## Figures and Tables

**Figure 1 ijerph-17-02118-f001:**
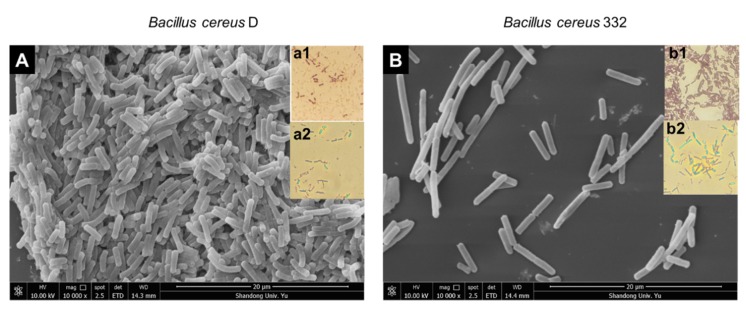
Micrographs showing the morphology of the *B. cereus* D (**A**) and *B. cereus 332* (**B**) strains observed by scanning electron microscopy, Gram-staining (a1) and spore-staining (a2) of *B. cereus* D, and Gram staining (b1) and spore staining (b2) of *B. cereus* 332.

**Figure 2 ijerph-17-02118-f002:**
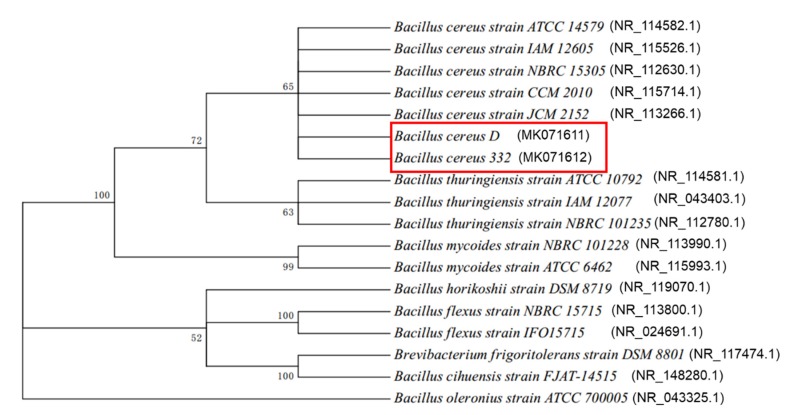
The phylogenetic tree generated from the 16S rDNA gene sequences of the two isolated strains, *B. cereus* D and *B. cereus* 332. Red block highlights the strains isolated in this study.

**Figure 3 ijerph-17-02118-f003:**
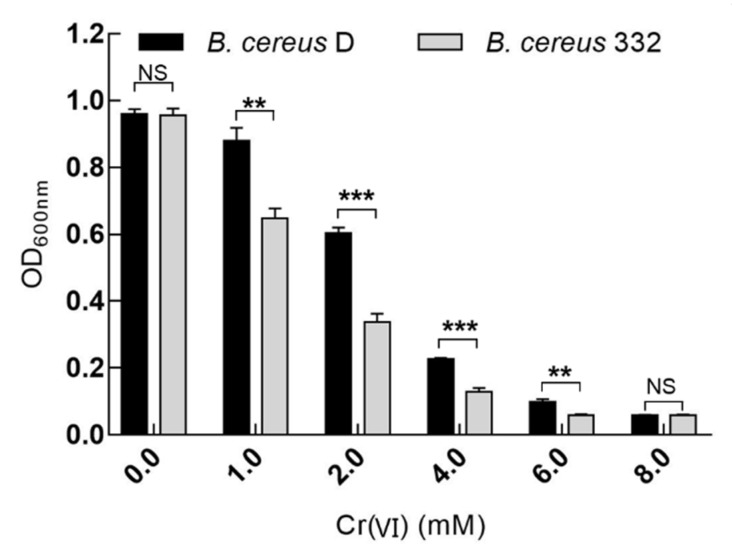
The effects of the concentration of Cr(VI) on the growth of *B. cereus* D and *B. cereus* 332. **** indicates *p* < 0.0001, *** indicates *p* < 0.001, ** indicates *p* < 0.01, and * indicates *p* < 0.05. NS indicates no significant difference.

**Figure 4 ijerph-17-02118-f004:**
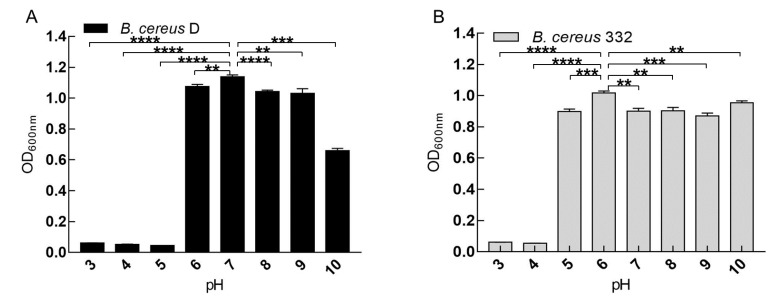
The effects of pH on the growth of *B. cereus* D (**A**) and *B. cereus* 332 (**B**). **** indicates *p* < 0.0001, *** indicates *p* < 0.001, ** indicates *p* < 0.01, and * indicates *p* < 0.05.

**Figure 5 ijerph-17-02118-f005:**
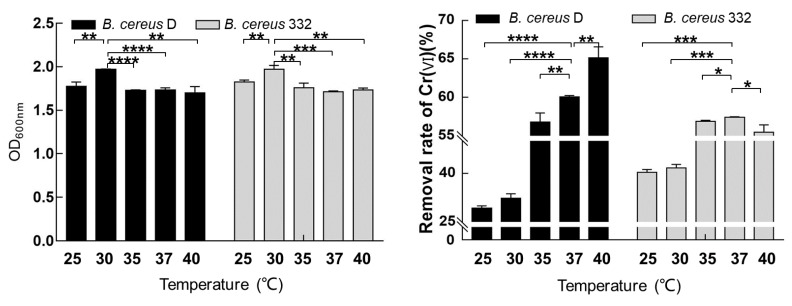
The effects of temperature on the growth (**A**) and the chromium-reducing capacity (**B**) of *B. cereus* D and *B. cereus* 332. **** indicates *p* < 0.0001, *** indicates *p* < 0.001, ** indicates *p* < 0.01, and * indicates *p* < 0.05.

**Figure 6 ijerph-17-02118-f006:**
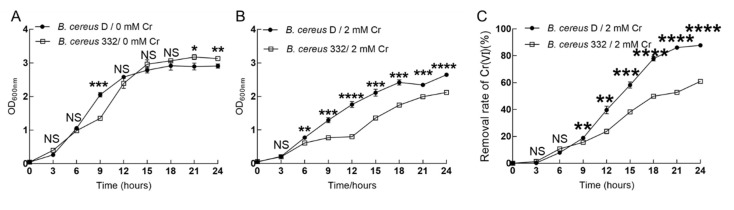
The growth of *B. cereus* D and *B. cereus* 332 without Cr(VI) (**A**) and with a 2 mM Cr(VI) concentration (**B**). The removal rate of Cr(VI) of *B. cereus* D and *B. cereus* 332 with a 2 mM Cr(VI) concentration (**C**). **** indicates *p* < 0.0001, *** indicates *p* < 0.001, ** indicates *p* < 0.01, and * indicates *p* < 0.05. NS indicates no significant difference.

**Figure 7 ijerph-17-02118-f007:**
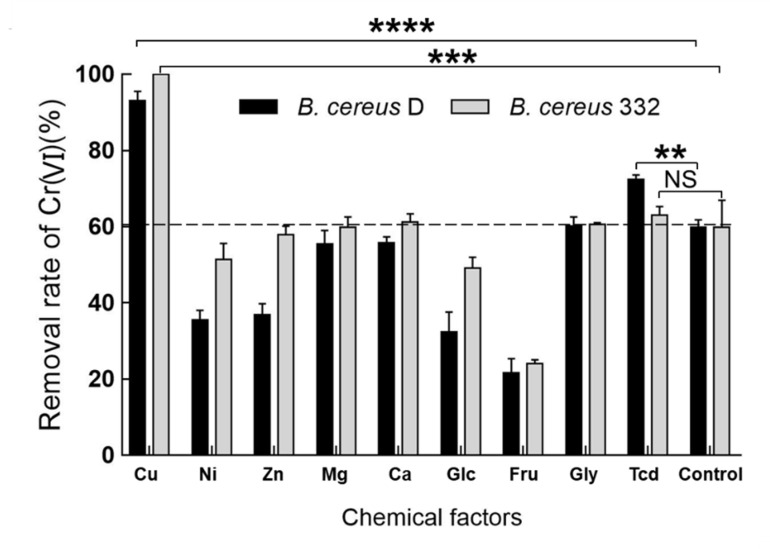
The effect of chemical factors (0.4 mM of Cu(II), Ni(II), Zn(II), Mn(II), and Ca(II), and 1 mM glucose (Glc), fructose (Fru), glycine (Gly), and trisodium citrate dihydrate (Tcd)) on the ability of *B. cereus* D and *B. cereus* 332 to reduce chromium at an initial concentration of 2 mM Cr(VI) at 37 °C, pH = 7.0. Control indicates without the addition of chemical factors. Significant difference analysis was only performed on the data that improved the chromium reduction ability. **** indicates *p* < 0.0001, *** indicates *p* < 0.001, ** indicates *p* < 0.01, NS indicates no significant difference.

**Figure 8 ijerph-17-02118-f008:**
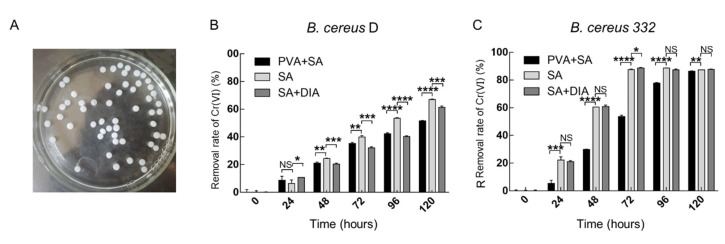
The morphology of immobilized cells (**A**). The Cr(VI) reduction in immobilized *B. cereus* D (**B**) and immobilized *B. cereus* 332 (**C**) with an initial 2 mM Cr(VI) concentration at 37 °C, pH = 7.0. PVA indicates polyvinyl alcohol, SA indicates sodium alginate, DIA indicates diatomite. **** indicates *p* < 0.0001, *** indicates *p* < 0.001, ** indicates *p* < 0.01, * indicates *p* < 0.05, NS indicates no significant difference.

**Table 1 ijerph-17-02118-t001:** The biochemical and physiological characteristics of *B. cereus* D and *B. cereus* 332.

Tests	Results		Tests	Results	
	*B. cereus* D	*B. cereus* 332		*B. cereus* D	*B. cereus* 332
**Morphological Tests**					
Clone shape	Round	Round	Pigment	-	-
Margin	Irregular	Irregular	Gram-reaction	+	+
Surface	Rough	Rough	Spore	+	+
Opacity	Opaque	Opaque	Cell shape	Rods	Rods
**Biochemical Tests**					
MR-VP test	+	+	Motive test	+	+
Lysozyme	+	+	Nitrate test	+	+
Lecithin	+	+	Acid production	+	+
Hemolysis	+	+	Protein crystal	-	-
Root shape growth	+	+	Casein decomposition	+	+

“+” indicates positive results and “-” indicates negative results.

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
