# Peer review of "Isolation and Identification of Chromium Reducing Bacillus Cereus Species from Chromium-Contaminated Soil for the Biological Detoxification of Chromium"

_ijerph, 2020, doi:10.3390/ijerph17062118_

Round 1
Reviewer 1 Report
As we all know, pollution and remediation is a hot issue all over the world. In this case, the paper focus on the chromium-contaminated soil. It is an interesting topic. But reading through the paper, I can not find the part related the soil. On the other hand, it is a big experiment report. So in my case, the comments should be supplied.
1) the abstract should be rewrite. It is hard to summary the paper.
2) The discussion and results should be resummary. In this case, it is poor.
3) SEM experiment looks nice, but two bacteria apperance are quite different or same. And the huge quantity difference reason?
4)There are so many conditional experiments, but I can not find the real summary.
5) On the background summary, about 30 references are listed. Is it necessary?
Anyway, there are big distance for publish in this journal.
Reviewer 2 Report
Isolation and identification of chromium reducing Bacillus cereus species for the bioremediation of chromium-contaminated soil.
In the above manuscript authors present two new isolates of B. cereus as potential sources of bioremediation for Cr(VI). Although it is well known that B. cereus species are tolerant to Cr(VI) and they are also able to reduce Cr(IV), it is useful to identify new isolates which would perform better to make the bioremediation of Cr(VI) more operationally and economically viable. Therefore, it is an important undertaking. However, I have the following major concern and few minor comments/ suggestions with ragrds to data discrepancies with in the figures, missing statistics and missing controls.
Major Concern: Discrepancy between data in Figures 3, 4 and 5
1.1. Growth of Isolates in 2 mM Cr(IV): Figure 5A shows that cereus D and B. cereus 332 grew to OD600 of 2.0 and almost 2.0 respectively in 24 hrs in a medium with Cr(VI) 2 mM at 37 ËšC at pH 7.0. However, in Figure 3C both the strains didn’t grow beyond OD600 = 1.0 even in Cr(VI) 0 (control) and with Cr(VI) 2 mM they grew to OD600 =~0.6 and ~0.4 (similar conditions as above: in 24 hrs at 37 ËšC at pH=7.0, Materials and Methods, Lines 117-121).
1.2. Ability of isolates to reduce Cr(VI): Similarly, Figure 5B shows that cereus D and B. cereus 332 removed ~85% and ~66% of Cr(VI) respectively when grown in 2 mM Cr(VI) at 37 ËšC, pH7.0. However, Figure 4 shows ~60% and ~<60% removal of Cr(VI) for the strains (similar conditions as above: in 24 hrs at 37 ËšC at pH=7.0, Materials and Methods, Lines 122-125).
Which data set is representative?
Minor Concerns / suggestions
- Except for Figure 5 statistical significance is not shown (tested?) for any other Figures. Nevertheless, authors discuss significance when comparing different conditions with regards to Figures 3, 4, 6 and 7 in the text. Statistically significance should be clearly shown in figures with information in figure captions.
- Figure 3C shows that the two isolates grown poorly in Cr(VI) above concentrations of 2 mM. However, authors claim (Lines 185-186) “As shown in Figure. 3C the two tested isolates survived under 8 mM Cr(VI) and have grown well in 4 mM Cr(VI)”. According to Figure 3C both strains show more than 75% reduction in growth in 4 mM Cr(VI). Therefore, this is an exaggeration and need to be corrected.
- No control was included in Figure 6. It is important to include a control without any supplements to compared to.
4. Suggestion: Simple t-tests may suffice, and it is generally acceptable to look at one variable at a time. It would have been better to use multiple linear regression to model the performance of strains under the different parameters tested. It may help to make a more informed decision about optimal conditions for industrial use of these species.
Reviewer 3 Report
- the bacterial species should be written in italic font in the title
- you may consider add more literature review to the introduction
- methods should have been explained in more detailed approach
- you could have been elaborated more on your results
Round 2
Reviewer 1 Report
In this version, it looks like better. All the comments are corrected nice. In this type, it is useful for readers.
Author Response
Point 1: In this version, it looks like better. All the comments are corrected nice. In this type, it is useful for readers.
Response 1: Thank you for your careful and valuable comments.
Reviewer 2 Report
The title has a typo "detoxication"
I did not understand why the authors reshufflled many figures?
Author Response
Point 1: The title has a typo "detoxication".
Response 1: Thank you for your careful comments. The typo was corrected in the new manuscript and was highlighted with a yellow shadow (Line 4, 63, 325).
Point 2: I did not understand why the authors reshuffled many figures?
Response 2: Thank you for your comments. The resubmitted manuscript was revised according to the suggestions of the reviewers. And we reshuffled the figures to highlight the research theme. The figures presentation of the new manuscript are more conducive to show our results and research significance.